# Alternative polyadenylation diversifies post-transcriptional regulation by selective RNA–protein interactions

Ishaan Gupta[1], Sandra Clauder-Münster[1], Bernd Klaus[2], Aino I Järvelin[1], Raeka S Aiyar[1], Vladimir Benes[3], Stefan Wilkening[1,4], Wolfgang Huber[1], Vicent Pelechano[1,*] & Lars M Steinmetz[1,5,6,**]

## Abstract

Recent research has uncovered extensive variability in the boundaries of transcript isoforms, yet the functional consequences of this variation remain largely unexplored. Here, we systematically discriminate between the molecular phenotypes of overlapping coding and non-coding transcriptional events from each genic locus using a novel genome-wide, nucleotide-resolution technique to quantify the half-lives of 3′ transcript isoforms in yeast. Our results reveal widespread differences in stability among isoforms for hundreds of genes in a single condition, and that variation of even a single nucleotide in the 3′ untranslated region (UTR) can affect transcript stability. While previous instances of negative associations between 3′ UTR length and transcript stability have been reported, here, we find that shorter isoforms are not necessarily more stable. We demonstrate the role of RNA-protein interactions in conditioning isoform-specific stability, showing that PUF3 binds and destabilizes specific polyadenylation isoforms. Our findings indicate that although the functional elements of a gene are encoded in DNA sequence, the selective incorporation of these elements into RNA through transcript boundary variation allows a single gene to have diverse functional consequences.

**Keywords** alternative polyadenylation; RNA stability; RNA-binding protein; transcript isoforms; 3′UTR
**Subject Categories** Genome-Scale & Integrative Biology; Transcription
**Mol Syst Biol. (2014) 10: 719**

## Introduction

Production of RNAs executes the genetic program encoded in genomic sequences in a temporally and spatially controlled manner. This control is exerted at multiple stages of gene expression regulation, from chromatin remodeling and transcriptional initiation to post-transcriptional regulation of RNA stability and translation. The post-transcriptional fate of an RNA molecule is closely linked to its transcription via co-transcriptional modifications and factors loaded onto the RNA during transcription that regulate RNA stability and processivity (Mata *et al*, 2005; Besse & Ephrussi, 2008; Belasco, 2010). In addition, the untranslated region at the 3′ end of transcripts (3′UTR) harbors many sequence elements that are targeted by post-transcriptional regulators such as RNA-binding proteins (RBPs) and miRNAs that influence mRNA stability, localization, and translation (Elkon *et al*, 2013). The position of polyadenylation thus determines whether these factors governing post-transcriptional fate are encoded in a transcript.

Recent protocols designed to efficiently map polyadenylation events have revealed that genes produce several 3′ isoforms, many of which co-exist even within the same clonal population of cells (Ozsolak *et al*, 2010; Shepard *et al*, 2011; Derti *et al*, 2012; Pelechano *et al*, 2013; Wilkening *et al*, 2013). Approximately 75% of human genes (Elkon *et al*, 2013) and almost all yeast genes express alternative 3′ isoforms (Moqtaderi *et al*, 2013; Pelechano *et al*, 2013; Wilkening *et al*, 2013). Besides the sheer diversity of 3′ isoforms across genes, the usage of 3′ isoforms is actively controlled during several biological processes. Tissue-specific usage of 3′ isoforms has been observed in humans (Derti *et al*, 2012) as well as in *Drosophila* (Smibert *et al*, 2012), where cells in the nervous system tend to have the longest 3′ isoforms, while cells in the gonads tend to have the shortest 3′ isoforms. Many studies have observed an anti-correlation between the lengths of 3′ isoforms and the

1  European Molecular Biology Laboratory (EMBL), Genome Biology Unit, Heidelberg, Germany
2  European Molecular Biology Laboratory (EMBL), Centre for Statistical Data Analysis, Heidelberg, Germany
3  European Molecular Biology Laboratory (EMBL), Genomics Core Facility, Heidelberg, Germany
4  Department of Translational Oncology, National Center for Tumor Diseases (NCT), Heidelberg, Germany
5  Department of Genetics, Stanford University School of Medicine, Stanford, CA, USA
6  Stanford Genome Technology Center, Palo Alto, CA, USA
   *Corresponding author: Tel: +49 6221 387 8389; Fax: +49 6221 387 8518; E-mail: pelechan@embl.de
   **Corresponding author: Tel: +49 6221 387 8542; Fax: +49 6221 387 8518; E-mail: larsms@embl.de

proliferative state of the cells (Sandberg *et al*, 2008; Lin *et al*, 2012). The 3′ isoforms are also regulated dynamically in response to stimuli, as shown in unicellular yeast (Yoon & Brem, 2010) and neurons of the multicellular rat (Flavell *et al*, 2008). Most such studies have focused on variation across different cell types or conditions; yet, the transcript heterogeneity present even within a single environmental condition suggests that every cell in a clonal population can have a unique transcriptome in terms of regulatory potential that could increase cell-to-cell heterogeneity and potentially confer evolutionary advantages (Pelechano *et al*, 2013).

Although many studies have described the extensive modulation of transcript isoforms, very few have explored the mechanisms by which different 3′ isoforms interact with cellular machinery to produce a molecular phenotype, such as altered translation efficiency or transcript stability. Some studies have shown that the production of shorter 3′ isoforms leads to higher translation efficiency due to the exclusion of miRNA-binding sites present in the longer isoforms (Sandberg *et al*, 2008; Mayr & Bartel, 2009). Another study has demonstrated that the RNA-binding protein (RBP) HuD regulates the stability of the gene coding for the brain-derived neurotrophic factor in an isoform-specific manner (Allen *et al*, 2013). RBPs can act as regulators of gene expression, controlling post-transcriptional processes and often binding to 3′ UTRs (Ray *et al*, 2013). It has been shown that 3′ UTRs of yeast genes harbor several RBP motifs (Hogan *et al*, 2008) that regulate their stability through RBPs such as the PUF proteins (Ulbricht & Olivas, 2008). In fact, a recent study in murine cells identified the motif for PUF RBPs as the most likely cause of the destabilization of long 3′ isoforms (Spies *et al*, 2013). Notably, the regions of UTRs that vary due to alternative isoform usage are statistically enriched in RBP motifs (Pelechano *et al*, 2013). Despite the wealth of knowledge on both alternative polyadenylation and RBPs, no study thus far has explored how the stability of 3′ isoforms can be shaped genome-wide by their interactions with particular RBPs. In addition, the sufficiently precise measurement of differences in RNA stability remains a technical challenge, which has in turn made it challenging to determine the contribution of alternative polyadenylation to these differences. Given the potential of 3′ isoform variation to impact inclusion of regulatory elements within 3′UTRs and thus post- transcriptional processing, there is a need to develop integrated methodologies for characterizing isoform function.

In this study, to investigate the post-transcriptional implications of isoform variation, we characterized the turnover of 3′ transcript isoforms in budding yeast *Saccharomyces cerevisiae*. For this, we developed an approach to determine decay rates of individual 3′ end isoforms at single-nucleotide resolution for the entire transcriptome. The approach is easily adaptable to other organisms. Our findings demonstrate that alternative isoforms differing in only a few nucleotides at the 3′ end of a transcript can have up to fourfold differences in their decay rates. Additionally, we distinguished between the contribution of coding and non-coding transcripts to the transcriptional output from a locus by measuring the turnover of overlapping transcripts along the yeast genome. To understand the contribution of RNA-binding proteins (RBPs) to regulating stability of specific coding transcripts of a gene, we also developed an assay to detect physical interactions between RBPs and specific 3′ isoforms across the entire yeast genome. With these approaches, we showed that the association of specific RBPs such as PUF3 with particular 3′ isoforms

of a gene regulates their decay rates. Finally, we validated the causative role of PUF3 in these interactions to modulate decay rates in an isoform-specific manner. Our results illustrate that the decay rate of each gene product is a composite of the decay rates from individual isoforms. They also reveal that alternative polyadenylation provides a means for the diversification of post-transcriptional phenotypes.

# Results

## Measurement of decay rates of 3′ isoforms

To robustly measure decay rates of RNA 3′ isoforms genome-wide, we established a novel high-throughput sequencing method called MIST-Seq (Measurement of Isoform-Specific Turnover using Sequencing). The method is based on accurate quantification of a decaying population of RNA and estimation of decay rates accounting for technical noise in RNA sequencing (Supplementary Information). We implemented MIST-Seq in a temperature-sensitive strain of *Saccharomyces cerevisiae* containing a modified *rpb1-1* allele that arrests RNA polymerase II transcription at 37°C (Nonet *et al*, 1987; Grigull *et al*, 2004). We chose this approach over alternative approaches of metabolic labeling, as it requires no assumptions about transcript length (Miller *et al*, 2011). We then used our previously developed 3′ T-fill sequencing approach (Wilkening *et al*, 2013) to quantify 3′ polyadenylation isoforms after shifting an exponentially growing culture from 24 to 37°C and profiling the transcriptome at 5, 10, 20, and 40 min after transcriptional arrest (Fig 1A). We restricted the time course to 40 min, since it has been reported that rRNA and protein production begin to deviate from the steady state 1 h after transcriptional arrest, affecting global metabolism and possibly leading to non-physiological measurements of decay rates (Nonet *et al*, 1987). This consideration also deterred us from using less invasive methods of transcriptional arrest such as "anchor-away", which inhibits transcription by exporting the polymerase outside the nucleus, but requires about 40 min to completely remove polymerase from highly transcribed genes (Fan *et al*, 2011). To ensure that our quantifications of RNA decay rates were accurate, we spiked in an equal amount of foreign RNA into the total RNA at each time point (1 μg *S. pombe* for every 9 μg of *S. cerevisiae*), similarly to a previous approach using microarrays (Sun *et al*, 2012). As the population of reads mapping to *S. pombe* represented the same number of transcript molecules at each time point, we could use these counts to calibrate the read counts obtained from the *S. cerevisiae* transcriptome and measure the decay of *S. cerevisiae* polyadenylated RNAs across subsequent time points (Fig 1B and C).

Accurate measurement of decay rates has proven challenging in previous studies, as evidenced by poor correlation of decay rates between experiments carried out in different laboratories, even when using the same experimental setup (Grigull *et al*, 2004). To ensure biological reproducibility, we carried out experiments using two independent biological replicates. We also accounted for the fact that mRNA sequencing data display higher relative variability of measurements of low expression levels (Anders & Huber, 2010). This variability becomes a concern during transcriptional arrest, since at later time points the number of counts per isoform decreases (Fig 2A and Supplementary Fig S1A). We found that

                                                                            

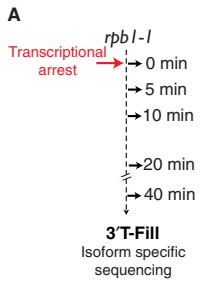
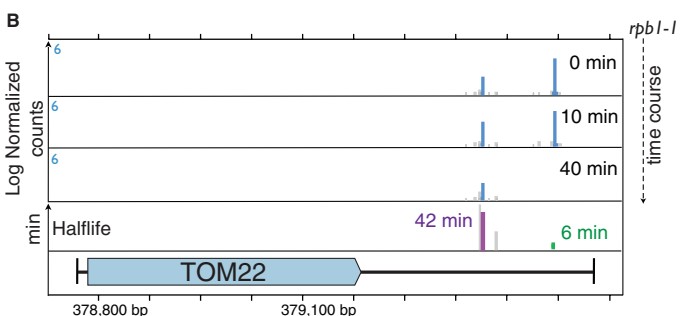
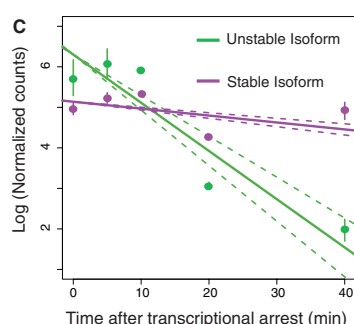

**Figure 1.   Isoform-specific measurement of transcript decay rates.**

A     Outline of MIST-Seq protocol to measure isoform-specific decay rates. Exponentially growing cells are shifted to a restrictive temperature that causes transcriptional arrest in the polymerase mutant *rpb1-1*. Total RNA is extracted at multiple time points, mixed with equal amounts of *S. pombe* RNA, and sequenced using 3′ T-fill to identify transcript 3′ ends at nucleotide resolution.

B     Snapshot of TOM22 isoform stability data. Normalized counts for 3′ isoforms of the gene across three time points (blue) and their corresponding half-lives (green and purple) are displayed. Only decay rates for 3′ isoforms with reliable quantifications are shown.

C     Log fit (solid lines, deviation in fit in dashed line) to the decreasing number of normalized read counts for the isoforms displayed in (B) in green and purple demonstrates the decay rate of the respective isoforms.

failure to take this variability into account may lead to overestimation of decay rates, especially for transcripts that decay rapidly (Fig 2B and Supplementary Fig S1B). We addressed this issue by using a weighted linear regression that accounts for intensity-dependent precision when estimating decay rates (Fig 2B) (detailed in Supplementary Information).

We obtained robust measurements of decay rates for approximately 21,600 isoforms at single-nucleotide resolution (without clustering or collapsing adjacent isoforms), mapping to 3,600 annotated protein coding and 210 annotated non-coding genes. The decay rates calculated from each replicate independently were in agreement (Spearman correlation 0.71, Fig 2C). We detected approximately 2,900 protein-coding genes with at least two 3′ isoforms mapping within the annotated coding sequence (CDS) or the 3′ UTR of the gene (Xu *et al*, 2009). The measurements of decay rates for individual genes agreed reasonably well with previous per-gene measurements of decay rates obtained using microarrays in the same yeast strain and a similar experimental setup (Spearman correlation 0.51, Supplementary Fig S1C) (Wang *et al*, 2002). We also confirmed the isoform-specific difference in decay rates for specific genes using RT-qPCR (Supplementary Fig S2). The 3′ isoforms had a wide range of decay rates, corresponding to half-lives from 4 to 180 min. The median half-life of the captured mRNA transcriptome was 19 min (Supplementary Fig S1C), in agreement with previous experiments (Wang *et al*, 2002; Grigull *et al*, 2004). It has been observed that the decay rates of transcripts involved in the same biological processes are correlated (Wang *et al*, 2002). Our data displayed similar trends at the level of decay rates for 3′ isoforms of genes with matching Gene Ontology functional annotations (Supplementary Table S1).

**Relationship between full-length isoform structure and decay rate**

A polyadenylation event provides information on the 3′ UTR of a transcript, but this clarifies neither its full structure nor whether the isoform covers the entire coding sequence (CDS). The latter information is important for precisely investigating the

post-transcriptional fate of a gene product. To circumvent this issue and investigate the relationship between coding potential and decay rate of 3′ isoforms, we mapped the detected 3′ isoforms in our study to a genome-wide annotation of full-length transcript isoforms that we previously generated (Pelechano *et al*, 2013). We thus assigned 65% of our 3′ isoforms to an annotated 5′ end, the remainder likely unmapped due to higher coverage in the current study. Consequently, each 3′ end mapped to one or more 5′ ends and we categorized the 3′ isoforms according to whether they covered the entire CDS of a gene. Isoforms that completely covered the CDS were classified as coding isoforms. All other isoforms – further classified as those mapping to the UTRs, intergenic regions, or only partially covering a translated region – were classified as non-coding isoforms.

As a result of this classification, we were able to gain insights into the transcriptional turnover of the entire yeast genome. We found that the coding isoforms had significantly lower decay rates than non-coding isoforms, decaying 23% slower ($P < 2.2e-16$; with median half-lives of 19 min for coding isoforms and 14 min for non-coding ones). Furthermore, we leveraged the full-length isoform annotations to disentangle the assignment of coding and non-coding isoforms to various polyadenylation events within annotated genes (Fig 3). We found that polyadenylation events in the 3′ UTR could belong to either coding or non-coding isoforms (Supplementary Fig S3A), which affected their stability: non-coding isoforms decayed 38% faster than coding isoforms (Fig 4, Supplementary Table S2). It is therefore essential to distinguish between coding and non-coding isoforms of a gene to correctly assess molecular phenotypes such as decay rates of coding gene products. Altogether we defined eight categories of transcript isoforms according to their coverage of annotated genes (Fig 4A). To estimate the statistical significance of the differences in decay rates between transcript isoform categories, we performed an overall ANOVA followed by pairwise t-tests (Fig 4B and Supplementary Table S2). Among the coding isoforms, we found that bicistronic isoforms covering two or more CDS were more stable than coding isoforms covering only a single CDS. Among the non-coding transcript isoforms, we measured for the first time the decay rates for 26 isoforms mapping to Stable Unannotated Transcripts or SUTs (Xu *et al*, 2009) and found them to be far

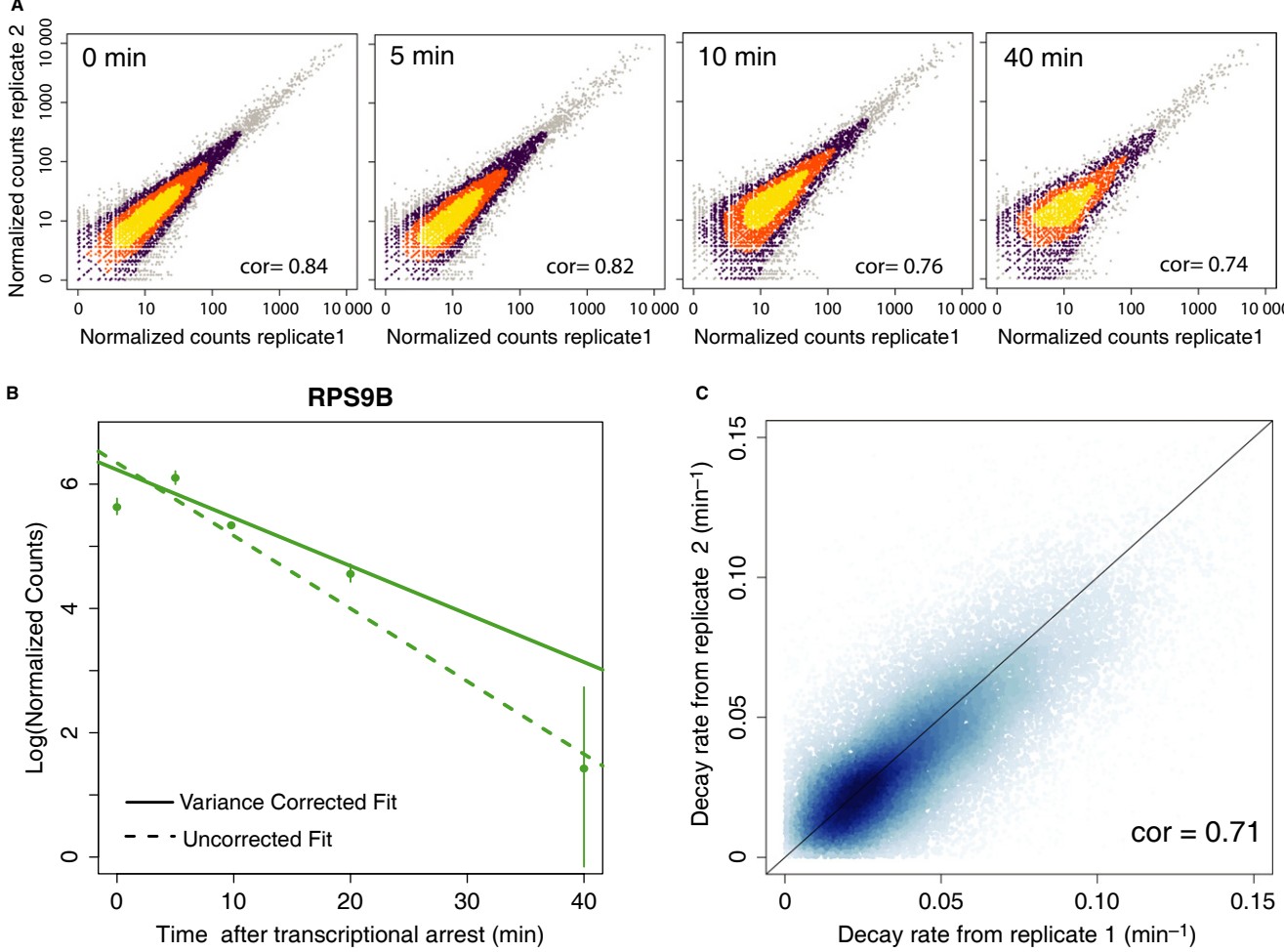

**Figure 2.    Accurate quantification of decay rates enabled by the use of weighted regression.**

A    Normalized sequencing counts per isoform from two biological replicates across the time points. Higher data density is represented by brighter colors. The increasing spread of counts at later time points (reflected by the decreasing Spearman correlation, cor) suggests an increase in the technical noise and the need for a weighted fitted scheme (see Supplementary Information).

B    Estimation of decay rates for isoforms of the gene *RPS9B* using traditional regression (dashed line) vs. the variance-corrected weighted regression used in this study (solid lines). The variance-corrected fit results in a more conservative estimate of decay rates, while the uncorrected fit ignores the increased noise due to lower read counts typically observed at later timepoints and is thus more error-prone (see Supplementary Fig S1B for more examples).

C    Reproducibility of all 3′ isoforms mapped to genes in the yeast genome across both biological replicates. Higher data density is represented by darker colors.

less stable than coding transcripts, with an median half-life of 14 min (as compared to 19 min for coding transcripts, although this trend is not statistically significant due to the small number of instances of SUTs). In contrast, we found that both intragenic and intergenic isoforms displayed decay rates similar to coding isoforms. Notably, we found that isoforms that partially covered translated portions of two adjacent genes were the least stable of all transcript classes, with decay rates more than twofold above the global average (with half-lives of 11 min; Fig 4). It has been suggested that the decay rates of divergent genes that share a common promoter region are weakly correlated (Dori-Bachash *et al*, 2012). We did not observe this correlation between the coding isoforms from divergent gene pairs firing bidirectionally from within 200 bp of each other (Supplementary Fig S4).

In summary, by combining our robust measurements of RNA stability with a dataset of joint annotation of transcript 5′ and 3′

boundaries (Pelechano *et al*, 2013), we disentangled the contribution of coding and non-coding isoforms to the turnover of transcription from each locus in the yeast genome.

## One gene, multiple decay rates

To explore the effect of alternative 3′ isoform usage on the decay rates of translated mRNAs, we restricted subsequent analyses to coding isoforms, which mapped to 3600 protein-coding genes. We detected 550 genes (15% of 3600) in which 3′ isoforms displayed significant differences in decay rates (FDR < 0.1) (Supplementary Table S3). We found that 244 of these genes expressed 3′ isoforms whose decay rates varied more than fourfold. Additionally, a considerable fraction of transcript molecules per gene display differing stabilities (Supplementary Fig S3B and C). Although differences in the 5′ UTR may also contribute to RNA decay (Trcek *et al*, 2011),

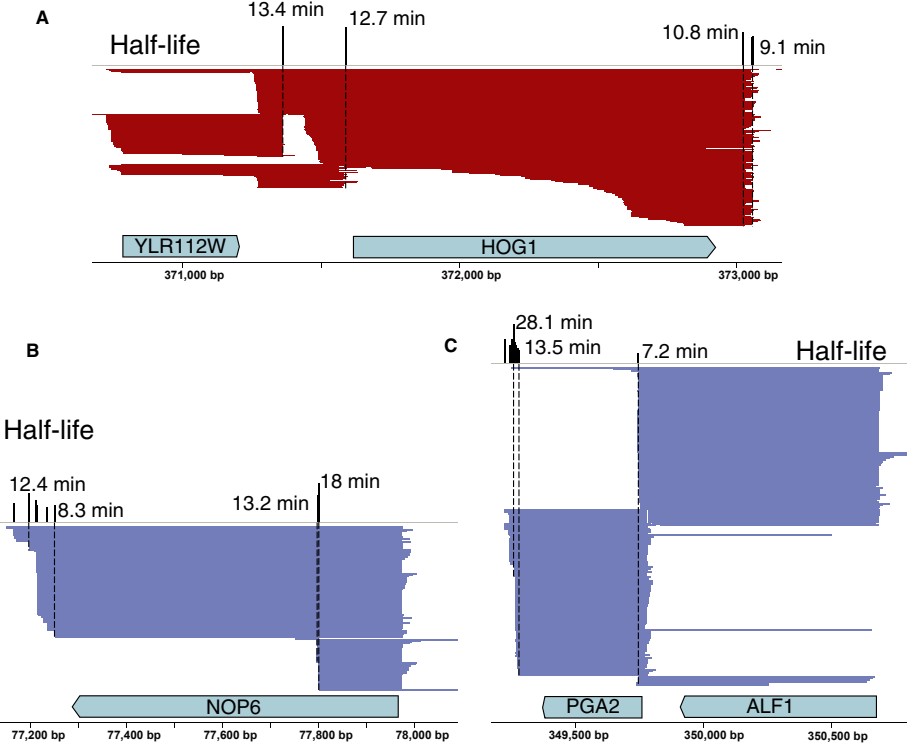

**Figure 3. Assigning 3′ isoform stability to full-length isoforms accounts for overlapping transcriptional architecture.**

A–C  Examples of genes with varying transcriptional architectures including: (A) complex locus, (B) internal termination events, (C) tandem overlapping isoforms. Half-lives of 3′ isoforms are shown in black above the uniquely mapped full-length transcript isoforms from (Pelechano *et al*, 2013) in red and blue (plus and minus strand).

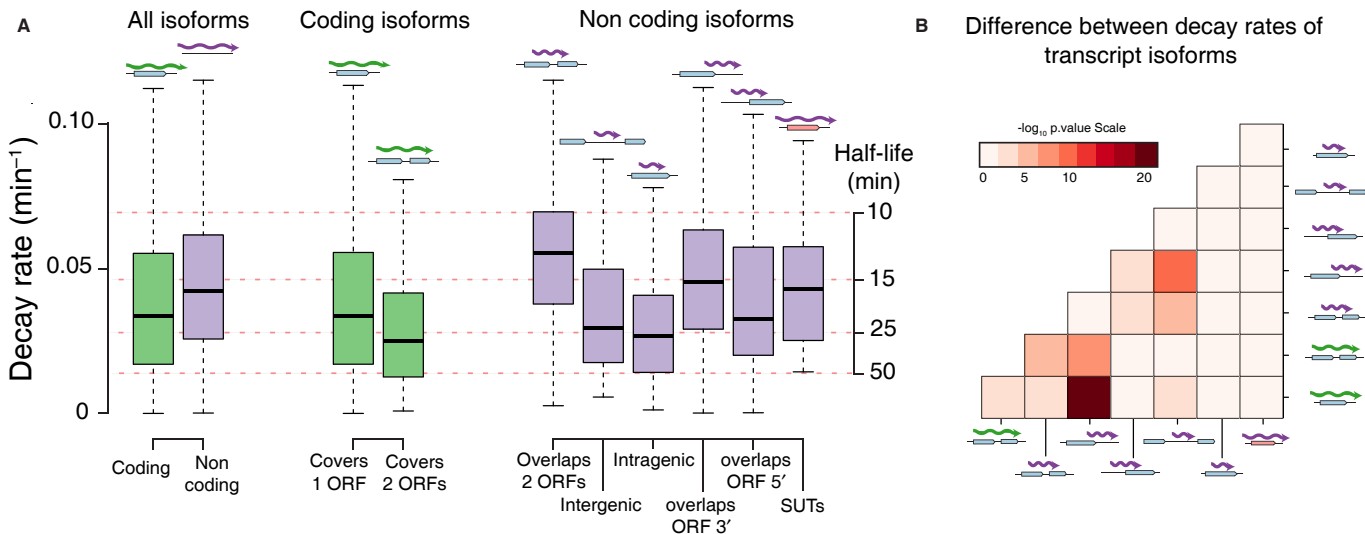

**Figure 4. Transcript isoform stability varies according to coding potential and genomic organization.**

A  Boxplots for the decay rates of different categories of transcript isoforms are displayed. Coding isoforms (green) are generally more stable than non-coding isoforms (purple). Isoforms are further classified into coding (covering 1 ORFs, or 2 or more ORFs) and non-coding (overlapping but not completely covering 2 ORFs, intragenic, overlapping the 3′ or 5′ of an ORF, or assigned to a stable unannotated transcript (SUT)). Only categories containing at least 20 unique instances are displayed.

B  Results from a statistical test (overall ANOVA followed by pair-wise *t*-test) for the decay rates between different categories of transcript isoforms (numerical results can be found in Supplementary Table S2)

our results demonstrate genome-wide that multiple mRNA messages encoding the same protein-coding sequence that differ only in their 3′ UTR can have divergent post-transcriptional lifetimes. This finding is in agreement with previous reports describing the potential role of the 3′ UTR in regulating stability and represents the first such genome-wide confirmation in yeast (Goodarzi *et al*, 2012; Allen *et al*, 2013; Ray *et al*, 2013).

### A shorter 3′UTR does not imply greater stability

Studies in higher eukaryotes have shown that a change in 3′ UTR length can exclude sequence motifs that facilitate decay and thereby stabilize shortened 3′ isoforms for a few genes across several cancer cell lines (Mayr & Bartel, 2009; Lin *et al*, 2012). As described above, a recent study has characterized the effect of the use of alternative 3′ UTRs on the decay rates of proximal and distal isoforms in murine cells to find only a slight destabilization of longer isoforms (Spies *et al*, 2013). Since our study detected numerous intermediate isoforms, with approximately 1,900 genes expressing three or more coding 3′ isoforms, we quantitatively investigated the association between 3′ UTR length and stability. Using our stringent stability measures, we observed no correlation between mRNA decay rate and 3′UTR length when considering all coding isoforms (Fig 5A). This still holds true when considering only the isoforms with differential decay rates for each gene (Fig 5B). These results suggest that, at least in *S. cerevisiae,* shorter 3′ UTRs do not always impart greater stability to transcripts.

To search for elements that govern 3′ isoform regulation, we restricted our analysis to about 6100 pairs of coding isoforms with statistically significant differences in decay rates (FDR < 0.1). We plotted the statistical confidence in calling differential stability between a pair of isoforms (represented by -log₁₀ p-value) against the distances between a pair of differentially decaying isoforms of the same gene (Fig 5C). Surprisingly, our results indicate that even a few nucleotide differences in 3′UTR length can lead to differential

RNA stability. In fact, we found 50 pairs of isoforms with significantly different half-lives separated by just a single nucleotide. The majority of isoforms with highly significant differences between their decay rates (*P* < 0.01) lie within 10–100 bp of each other. Among these 10–100 bp regions, we did not find any novel sequence motifs that could be mediating these changes in stability.

### RBPs contribute to isoform-specific decay rates

Previous studies have predicted 6–15 bp RBP target motifs in the 3′ UTR in *S. cerevisiae* (Riordan *et al*, 2011). Several of these RBPs, including PUF3, PUF4 (Ulbricht & Olivas, 2008) and VTS1 (Rendl *et al*, 2008), have been identified as regulators of mRNA stability. In addition, the variable 3′ UTR region among alternative 3′ isoforms is enriched for the presence of RBP motifs, highlighting their potential to modulate post-transcriptional fates of isoforms (Pelechano *et al*, 2013). To analyze the potential role of RBPs in isoform-specific RNA stability, we used a published set of RBP motifs (Riordan *et al*, 2011) and mapped the potential-binding sites of seven RBPs to the yeast genome. In the case of PUF3, the presence of the motif was sufficient to explain the observed differences in decay rates; there was no effect of the distance between the motif site and the 3′ end of the isoform (Fig 6A) on decay rates. Therefore, from genes with predicted RBP motifs, we separated the isoforms with and without the motif and calculated cumulative decay rates in these two classes for each of the seven RBPs (Fig 6B). This analysis suggested that RBPs regulate the stability of genes in an isoform-specific manner (Supplementary Table S4). In particular, isoforms containing the PUF3 motif were markedly destabilized. Although the effects of RBPs on mediating RNA stability have been extensively studied, it has neither been established genome-wide whether RBPs interact with transcripts in an isoform-specific manner, nor how these interactions shape the stability of the transcriptome. To demonstrate this, we used PUF3 as a proof of concept to validate that the stability

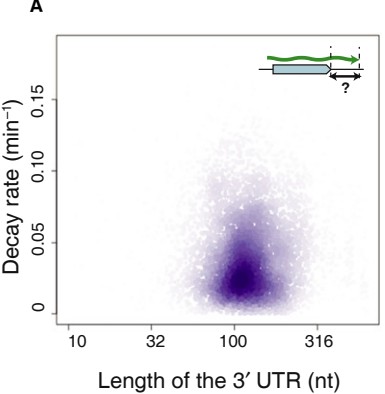
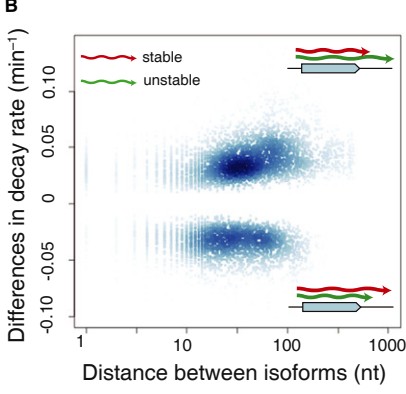
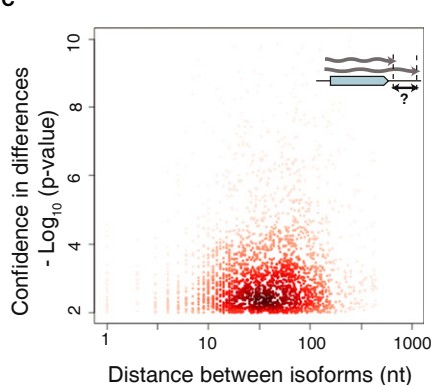

**Figure 5.  UTR length is not associated with decay rates.**

A    Scatterplot of decay rates (*y*-axis) vs. the length of the 3′UTR on (*x*-axis) reveals no direct correlation between 3′UTR length and isoform stability in the yeast transcriptome.

B    Differential decay rates between pairs of proximal and distal isoforms (*y*-axis) shows that UTR length differences (*x*-axis) are associated with both increased and decreased stabilities (top and bottom respectively).

C    Most pairs of isoforms with significant stability differences (*y*-axis) lie within 10–100 bp of each other (*x*-axis).

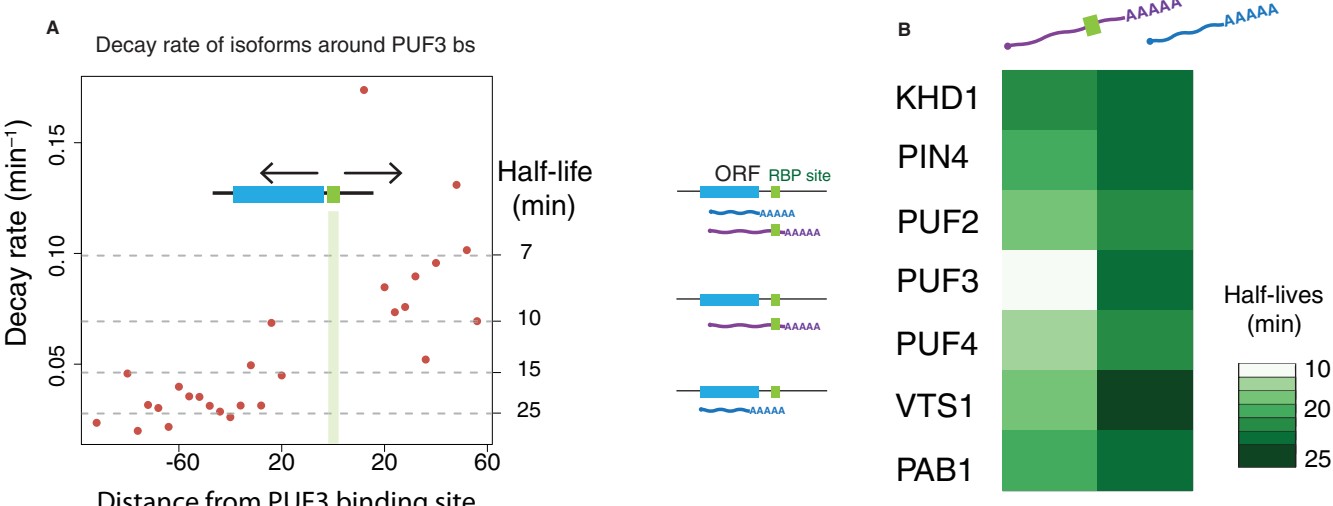

**Figure 6.   RNA-binding proteins (RBPs) contribute to isoform-specific decay rates.**

A   The presence of a PUF3 binding motif is associated with a sharp decrease in isoform stability. Decay rate (*y*-axis) and distance from the annotated PUF3 binding site (*x*-axis) for those genes with annotated PUF3 binding sites in their 3′UTRs are shown.

B   Isoform stabilities classified according to the presence (purple) or absence (green) of each of the 13 RBP motifs from (Riordan *et al*, 2011). The presence vs. absence of PUF3 motifs is associated with the greatest difference in isoform half-lives.

of transcript isoforms can be modulated by their selective interaction with RNA-binding proteins.

### RBPs can selectively bind to different 3′ isoforms of a gene

Most previous studies have relied on the presence of predicted RBP motifs in transcript sequences. To obtain direct biochemical evidence for the selective interaction of RBPs with individual 3′ isoforms, our study established a method to measure isoform-specific protein binding (isRIP, for isoform-specific RNA immuno-precipitation). The method is based on accurate quantification of isoforms that were immunoprecipitated with an RBP. We used a TAP-tagged yeast strain (Gavin *et al*, 2002) and immunoprecipitated RNA natively bound to the RBP without cross-linking the sample. We then applied genome-wide 3′ isoform sequencing (Wilkening *et al*, 2013) to measure the enrichment of 3′ isoforms in the immunoprecipitated (IP) fraction relative to the input sample in two biological replicates. We used the software '*DESeq2*' (Anders & Huber, 2010) to calculate normalized fold change in our IP signal to account for technical noise at low count values per isoform (Supplementary Fig S5).

We selected PUF3 as a candidate to perform is RIP because the presence of its binding motif was associated with the largest difference in decay rates in our analysis. PUF3 is a protein known to destabilize transcripts of nuclear-encoded mitochondrial genes in yeast (Olivas & Parker, 2000). In addition, a recent study in murine cells computationally predicted PUF motifs to be the strongest destabilizing regulators within isoforms (Spies *et al*, 2013). Gene-specific association of PUF3 has been demonstrated using RNA-IP on microarrays (Gerber *et al*, 2004), but the resolution of the arrays is not high enough to distinguish between isoforms. We applied isRIP to PUF3 and obtained a confident set of 73 3′ isoforms that bound to

the PUF3 protein with at least fourfold enrichment relative to input RNA (Fig 7A, Supplementary Table S5). Notably, PUF3 bound to both coding and non-coding isoforms (45 and 28 isoforms, respectively). The coding isoforms mapped to 44 protein-coding genes in the yeast transcriptome. Surprisingly, very few of these genes possess the predicted PUF3-binding sites (Riordan *et al*, 2011), underscoring the intrinsic limitations of motif prediction approaches. Among the isoforms from these 44 genes, the average isRIP enrichment in the isoforms that were not bound by PUF3 was sixfold lower than the average enrichment of the isoforms associated with PUF3. We conclude that RBPs such as PUF3 can bind to transcripts in an isoform-specific manner.

### PUF3 regulates the decay rate of 3′ isoforms

Using the experimentally validated target isoforms of PUF3 obtained with isRIP, we tested whether PUF3 binding regulates isoform stability. The decay rates of the isoforms of the 44 genes identified above differed according to their potential to bind PUF3. Among these 44 genes, the average decay rate of the 45 coding isoforms that bound to PUF3 was twofold higher than the average decay rate for the 293 coding isoforms that were not bound to PUF3 (Fig 7B). This observation strongly indicates that PUF3 binds and destabilizes transcripts in an isoform-specific manner.

To confirm that PUF3 binding itself modulates transcript stability in an isoform-specific manner, we performed MIST-Seq in two biological replicates of a strain lacking PUF3 (*puf3Δ rpb1-1*) in the *rpb1-1* background (as for the wild-type strain). The median decay rate in the mutant strain was 10% higher than the median decay rate in the wild-type. This could be due to general RNA destabilization in the mutant strain or to the intrinsic technical variation of stability measurements. To circumvent these issues and disentangle

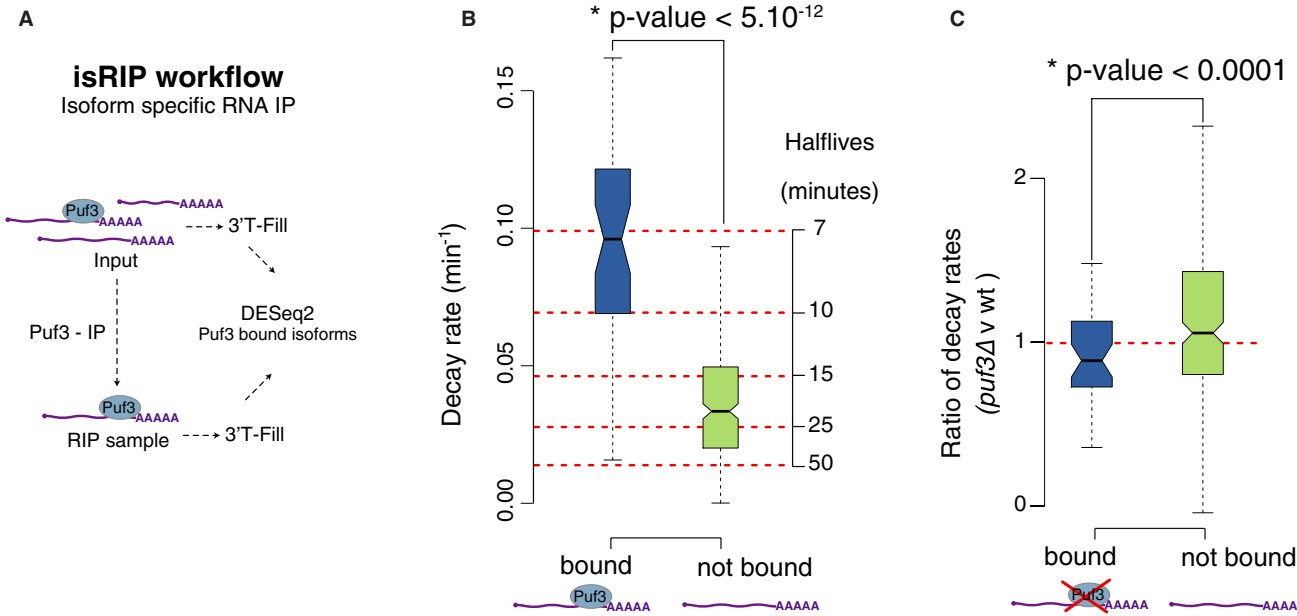

**Figure 7.  PUF3 selectively binds and destabilizes specific 3′ isoforms.**

A   Schematic of the isRIP protocol. We detected 73 bound isoforms (blue) that are enriched in the IP fraction at an FDR of 10% by at least four-fold vs. the input. Of these 73 isoforms, 45 have coding potential and map to 44 genes. The following plots display only the differences between the coding isoforms of these 44 genes.

B   The 45 coding isoforms bound by Puf3 (green) are 2.5 times less stable than the 293 isoforms not bound by Puf3 (blue).

C   Isoforms bound by PUF3 in WT (blue) have significantly lower relative decay rates (Ratio of decay rates per isoform in Δ*puf3* strain vs WT strain) than isoforms not bound to PUF3 in the wild-type (green), indicating a stabilization of the isoforms that bind to PUF3 upon its deletion. This demonstrates isoform specificity in the regulation of decay rates by PUF3.

the effects of PUF3 binding from the global shift in decay rate, we focused on the relative decay rates of each isoform, calculated as the ratio of the mutant to wild-type decay rates. We found that the decay rate ratios of the isoforms bound by PUF3 were significantly lower ($P$-value < 0.0001) than that of those not bound by PUF3 (Fig 7C). These findings confirm that isoforms bound to PUF3 in the wild-type are subject to PUF3-mediated decay, as evidenced by their higher decay rates. Interestingly, only coding isoforms bound by PUF3 differentially decayed in the *puf3*Δ strain, while the stability of non-coding isoforms bound by PUF3 was almost unaffected (Supplementary Fig S5). Our study thus demonstrates for the first time at a genome-wide level that alternative 3′ usage can result in isoform-specific targeting by RBPs, which in turn causes selective alterations in transcript isoform stability.

## Discussion

### Implications of isoform-specific turnover measurement

RNA turnover is an essential process that defines post-transcriptional behavior and diversifies the functional impact of each transcript. The turnover of a given transcript can be influenced by regulatory elements in its sequence (Goodarzi *et al*, 2012), yet most studies in this field have quantified only the cumulative RNA turnover within the boundaries of a gene without accounting for alternative isoforms. The results from our study provide the first genome-wide measurement of post-transcriptional properties of

alternative polyadenylation isoforms in *S. cerevisiae*. Using our new method MIST-Seq, we measured the decay rates of overlapping transcription events and characterized them according to their coding potential and genomic organization. We defined the coding potential of these isoforms using our previous full-length transcript isoform annotation (Pelechano *et al*, 2013). This enabled us to demonstrate the impact of alternative polyadenylation on the diversification of gene products. We found that alternative polyadenylation events separated by just a few nucleotides can lead to significant differences in stabilities, including 398 isoform pairs separated by less than 5 nucleotides (Supplementary Table S3). In some cases, the alternative binding of RBPs can explain the differences between stability of different isoforms of the same gene. In other cases, however, especially those in which only a single nucleotide confers differential stability, more research will be necessary to identify the responsible mechanisms and machinery. Differential loading of proteins during transcription termination, involvement of machinery detecting nonsense codons, or the effect of cotranscriptional folding of RNA could be contributing to stability differences between isoforms. Notably, these differences in stability can be substantial, as is the case for 244 genes for which alternative isoform stabilities vary over fourfold.

We quantified the decay rates of previously uncharacterized non-coding transcripts from intergenic and intragenic regions and found their decay rates to be similar to coding transcripts (Fig 3). Our study provides a measure of the turnover of stable unannotated transcripts (SUTs), most of which arise from bidirectional promoters of protein-coding genes (Xu *et al*, 2009). Most of the isoforms

mapping to SUTs were considerably less stable than coding transcripts. This relatively low stability of SUTs could be due to their association with components of the exosome machinery, as shown by a recent study using *in vivo* RNA-protein cross-linking (Schneider *et al*, 2012).

The measurement of decay rates of 3′ isoforms constitutes a new molecular phenotype with the potential for direct consequences on the cellular function of these transcripts. Such diversification of function through alternative 3′ isoforms has been demonstrated in a handful of studies in the form of altered protein abundance (Mayr & Bartel, 2009; Allen *et al*, 2013) or association with polyribosomes (Spies *et al*, 2013). Moreover, differential decay rates could also be biologically useful in response to stimuli. It has been shown that decay rates of genes encoding members of the same protein complex are similar (Wang *et al*, 2002) and that changes of their decay rates in response to stress can be coordinated (Miller *et al*, 2011). In our study, we accordingly observed similar decay rates for isoforms arising from genes in similar Gene Ontology functional categories (Supplementary Table S1). Our results suggest that the expression of particular polyadenylation isoforms across conditions could be used to regulate both decay rates and transcript abundance. Additionally, the expression of alternative isoforms even within a single environmental condition will diversify transcript stabilities and thus lead to increased cell-to-cell heterogeneity.

Our study demonstrates the breadth of isoform-dependent molecular phenotypes, even within a single environmental condition and genetic background. While the relative differences observed within a sample are likely independent of the specific method used to estimate RNA stability, the absolute calculations of transcript half-lives can vary significantly according to the method used (Miller *et al*, 2011; Sun *et al*, 2012). For example, the current implementation of MIST-Seq relies on a widely used method of transcriptional inhibition after heat shock in a temperature-sensitive strain (Nonet *et al*, 1987; Grigull *et al*, 2004). It is well known that this strategy elicits a stress response, and it should be noted that RNA stabilities are measured at 37°C. Other recently developed methods that rely on metabolic labeling of newly synthesized molecules followed by mathematical modeling (Miller *et al*, 2011) tend to estimate shorter half-lives than the *rpb1-1* approach. Such methodological differences should be accounted for when interpreting or comparing these datasets.

### Implications of isoform-specific binding of proteins

Coordination of decay rates can be achieved through common sequence elements encoded in the UTRs of genes (Goodarzi *et al*, 2012; Ray *et al*, 2013). In yeast, more than 70% of genes express alternative transcript isoforms with different combinations of RBP sites. Moreover, RBP sites are specifically enriched in the variable regions between isoforms, offering the potential to differentially regulate their post-transcriptional fate (Pelechano *et al*, 2013). Our results demonstrate that differences in the stabilities of 3′ isoforms are associated with the presence of RBP motifs. To gain further biochemical support for isoform-specific interaction of RBPs, we developed a new method named isRIP, which measures 3′ isoform-specific interactions with proteins. We applied this technique to PUF3, thereby experimentally validating a direct causal relationship between isoform-specific PUF3 binding and decay rate. We showed that for a given gene, only those isoforms bound by PUF3 are

destabilized. This demonstrates that different molecular phenotypes of 3′ isoforms such as decay rates may be exerted through their specific interactions with RBPs. Interestingly, we also found evidence of non-coding isoforms associated with RBPs other than the components of the RNA-processing machinery (Schneider *et al*, 2012). However, the specific binding of PUF3 to these non-coding isoforms did not alter their stability (Supplementary Fig S5C), suggesting that PUF3 (or the downstream degradation machinery) remains inactive on these substrates. One possibility is that the binding of these non-coding isoforms to PUF3 might play a role analogous to lncRNA decoys in higher eukaryotes that act as molecular sinks for RBPs (Wang & Chang, 2011) or sponges for miRNAs (Hansen *et al*, 2013). Further studies applying isoform-specific measurements of RBP binding will be necessary to characterize these interactions and their potential functional consequences.

Since transcripts often contain more than one RBP motif (Hogan *et al*, 2008), it is likely that combinations of RBPs associated with a particular UTR interact to produce specific phenotypic outcomes. Therefore, it will be necessary to consider both the existence of cis-regulatory elements in 3′ UTRs and the occurrence of these elements in transcript isoforms to improve our understanding of the post-transcriptional gene regulatory code. Extensive efforts are being made to discover more cis-regulatory elements by identifying new RBPs (Castello *et al*, 2013) and characterizing their binding sites (Hafner *et al*, 2010). Such studies are gaining momentum, especially in light of an increasing number of RBPs emerging as oncogenes (Spence *et al*, 2006) that exert effects on post-transcriptional phenotypes. The methods established in this study can be applied to investigate the regulatory potential of these new RBPs.

An open question is to what degree these post-transcriptional differences affect cellular heterogeneity. Notably, our study identified a few genes with two or more differentially stable 3′ isoforms that are expressed on average at less than one molecule per cell (Miura *et al*, 2008), meaning that any two cells in a clonal population may express isoforms of these genes with differing stabilities. More generally, our genome-wide observations suggest that genetically identical cells have heterogeneous transcriptomes with varying RNA stabilities and RBP interactions. Technological advances that enable quantification of full-length isoforms and their post-transcriptional properties will be important for defining the full functional impact of an expressed sequence. The use of less invasive and more versatile methods, such as metabolic labeling (Miller *et al*, 2011; Sun *et al*, 2012), which allows simultaneous measurements of transcriptional synthesis and decay, will enhance the functional characterization of transcript isoforms. Phenotypic characterization of transcriptional isoforms thus opens up an avenue to further our understanding of how information encoded in a genome is executed.

## Materials and Methods

### Sample preparation

We used *Saccharomyces cerevisiae* strain *rpb1-1* (Mat a, *his3Δ*, *leu2-3*, *ura3-52*, *rpb1*::HIS3 with plasmid RY2522) (Nonet *et al*, 1987) for transcriptional arrest, and the Puf3-TAP-tagged strain (BY4742 MAT α, *his3Δ1*, *leu2Δ0*, *lys2Δ0*, *ura3Δ0*, *pep4Δ*::KANR) (Gavin *et al*, 2002) for isRIP. Cells were grown to mid-log phase (OD$_{600}$~1) at

30°C using YPD (1% yeast extract, 2% peptone, 2% glucose). *Schizosaccharomyces pombe* (h-) was grown to mid-log phase (OD600~1) at 30°C using YES media (0.5% yeast extract, 3% glucose, supplemented with adenine, histidine, leucine, uracil and lysine).

## Data availability

The data described in this study are available from the ArrayExpress repository (www.ebi.ac.uk/arrayexpress) under the accession number [E-MTAB-2123].

## 3′T-fill sequencing and data analysis

The sequencing libraries were prepared as previously described (Wilkening *et al*, 2013). Sequencing reads were aligned to a composite genome of *S. cerevisiae* (version SGDR64), *S. pombe* (version 14), and *in vitro* transcripts (ATCC 87482, 87483 and 87484) spike-ins using the GSNAP aligner (Wu & Watanabe, 2005). The aligned reads were further filtered to remove potential false positive poly(A) site calls arising from internal mispriming as previously described (Wilkening *et al*, 2013). Poly(A) sites were assigned based on the first base of each read.

## MIST-Seq

For transcriptional arrest, the *rpb1-1* strain was grown at 24°C in two independent biological replicates, and the temperature was raised instantly to 37°C by the addition of warm media. At each point in the time course (0, 5, 10, 20, 40 mins), cells were harvested and flash-frozen in liquid nitrogen. Total RNA was isolated by phenol extraction with glass beads using a FastPrep-24 agitator (Zymo-research S6005) followed by phenol:chloroform:isoamyl alcohol cleaning and precipitation in ethanol.

## isRIP, isoform-specific RNA immunoprecipitation

Flash-frozen pellets of exponentially growing Puf3-TAP-tagged cells (OD$_{600}$ 0.5–0.8) were mechanically lysed with glass beads using a FastPrep24 agitator. An aliquot of each cell lysate was used as an input control. Cell lysates were subjected to immunoprecipitation with Dyna M280 sheep anti-rabbit IgG beads (Invitrogen) for 2 h at 4°C. RNA-protein complex was released by TEV protease cleavage of the TAP-tag. The input and immunoprecipitated RNA were purified by phenol:chloroform:isoamyl alcohol extraction. Experiments were performed in triplicate.

## Statistical procedures

*S. cerevisiae* read counts were normalized according to *S. pombe* reads and log-transformed. Decay rates per isoform were calculated by fitting a weighted linear regression to these transformed data. The weights were equal to the estimated variance of the log-counts, obtained by combining time point-specific dispersion estimates (Anders & Huber, 2010) with a Taylor approximation of the variance of the log-transformed counts.

For each isoform, these calculations resulted in an estimate of the decay rate and its standard error, from which we performed a two-sided *z*-test to call differential stability. In order to call

differential abundance between the input and the RNA-IP sample, we used the Bioconductor package DESeq2 (Anders & Huber, 2010) and defined differentially regulated isoforms as those that were detected at an FDR of 10% or less by the software, and had more than a fourfold change.

*P*-values reported in Fig 7 were determined by a *t*-test.

**Supplementary information** for this article is available online: http://msb.embopress.org

## Acknowledgements

We would like to thank J.E. Perez-Ortin (University of Valencia) and A.-C. Gavin (EMBL Heidelberg) for sharing yeast strains, S. Anders for assistance with the DESeq2 software, Charles Girardot for coordinating with ArrayExpress for data availability, as well as members of the Steinmetz lab, R. Bhardwaj and S. Tyagi for useful discussions. This study was technically supported by the EMBL Genomics Core Facility. This work was supported by grants from the Deutsche Forschungsgemeinschaft (1422/3-1) and National Institutes of Health to L.M.S.

## Author contributions

VP, IG and LMS designed the research; IG and BK analyzed the data with the help of AIJ.; IG, and SC performed the wet laboratory experiments with help from VP, SW and VB; LMS., VP, WH supervised the research; IG, RSA, VP, BK, AIJ and LMS wrote the manuscript.

## Conflict of interest

The authors declare that they have no conflict of interest.

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
