## [Review Process File · Molecular Systems Biology]

Alternative polyadenylation diversifies post-transcriptional regulation by selective RNA-protein interactions

Ishaan Gupta, Sandra Clauder-Münster, Bernd Klaus, Aino I Järvelin, Raeka S. Aiyar, Vladimir Benes, Stefan Wilkening, Wolfgang Huber, Vicent Pelechano and Lars M. Steinmetz

Corresponding author: Lars M. Steinmetz, European Molecular Biology Laboratory, Heidelberg

Review timeline:	Submission date:	17 December 2013
	Editorial Decision:	10 January 2014
	Revision received:	20 January 2014
	Editorial Decision:	22 January 2014
	Revision received:	26 January 2014
	Accepted:	27 January 2014

Editor: Thomas Lemberger

Transaction Report:

1st Editorial Decision

10 January 2014

Thank you again for submitting your work to Molecular Systems Biology.

We have now heard back from two of the three referees who accepted to evaluate the study. Given that their recommendations are very similar and in the interest of time, I prefer to make a decision now with the available reports. As you will see, the referees find the topic of your study of potential interest and are supportive. They raise however a series of concerns and make suggestions for modifications, which we would ask you to carefully address in a revision of the present work.

The methodology used in your study should thus be carefully discussed in the light of the alternative method of metabolic labeling.

One point that was not raised by these reviewers but that seems of importance is to provide information and discuss the abundance ratios of isoforms with different turnover rates.

Please include a 'Data availability' sub-section to Materials & Methods where you list the accession number of the datasets produced and analyzed in this study.

It is a great idea to include a Sweave .rnw document. Please include it within a zipped folder together with a README text file explaining how to compile it, list possible dependencies (R libraries) and what needs to be modified. If there is a way to make it generic (eg without reference to local directories and files), that would be great.

Reviewer #1:

The study by Gupta et al. investigates the influence of having transcript isoforms on transcript stability in the yeast *S. cerevisiae*. The main conclusion is that 3'-UTR heterogeneity can for some genes contribute to large differences in transcript stability and that in some cases this is mediated by differential interaction of 3'-UTRs with Puf3, an established regulator of mRNA stability. This is in part a confirmation of what is known/expected based on previous single gene and genome-wide transcript heterogeneity studies. The importance is the confirmation itself and the genome-wide scale of the study, that also allows an assessment of how widespread this type of regulation is. The data both serves as a demonstration that transcript isoforms can (in principle) contribute to phenotypes (indicating that this is not a particularly wide-spread phenomenon) and will also be useful for scientists wishing to investigate this for their particular favorite gene(s).

The general strategy used to determine decay rates is probably the one that has been most widely used in the past (shifting an RNA polymerase II temperature sensitive strain to the non-permissive temperature). This approach is a lot more problematic than the authors present. In the first place the allele already causes slower growth at the permissive temperature. Second, the measured decay rates are a combination of two things: the decay itself plus the effects of shifting the cells from 24 to 37 degrees. Recent work using an almost completely non-invasive method based on *in vivo* labeling of RNA with 4sU/4tU has quite convincingly shown that the invasive methods (eg Cu-phen, rpb1-1) are accompanied by massive, general effects on RNA stability due to the stress itself. It is really surprising that the authors don't use this approach (applied in yeast by the Cramer lab in a series of three papers by first authors Sun, Schwalb and Miller, some of which has been cited here) which is much, much better than rpb1-1 and yields quite different half-lives! Although this has an enormous impact on the decay rates determined here (essentially this all needs to be redone), this fortunately doesn't impact their general finding that different isoforms can influence stability. The authors should however properly explain the difference between rpb1-1 determined decay rates and those most recently determined using *in vivo* labeling, pointing out the large difference in determined values and that the *in vivo* labeling methods are superior because they are not invasive. It is really important to place the current study in the context of what has been done before and not to give the impression that using rpb1-1 is a good idea - it is no longer.

Figure 4 and the section describing these results. This figure should be accompanied by a statistical test (overall anova followed by pair-wise t test that is multiple testing corrected). The results of this test should also be clearly stated in the section describing this so that it is clear what the significance is of these different classes. If insignificant, this needs to be stated.

minor issues

page 4: "The decay rates for each independent replicate displayed good agreement (Spearman correlation 0.71, Figure 2C)". I would suggest removing the word "good" from this sentence. I guess it depends on what you are used to but a correlation of 0.71 between two replicates of the same measurement is actually quite bad.

typos

"after 1h after" (pg 4 line9)

"modulate" page 7 should be modulated

Reviewer #3:

Multiple studies have revealed high prevalence of alternative polyadenylation in every eukaryotic organism studied. Although numerous single genes studies have shown that transcripts with different cleavage and polyadenylation sites (PAS) behave differently, no genome-wide analysis has been performed to systematically analyse the behaviour of different isoforms. This manuscript addresses this issue by combining a high throughput method to identify PAS with a classic approach to measure RNA half-lives. This is a novel and clever approach to a very important question. Overall the paper is of good quality, well written and the data are carefully interpreted. I do not think that the choice of method for measuring RNA stability is the best (see below), but this is still very good piece of work. I only have a few minor comments and remarks:

1. I did not see any information on data deposition, all sequencing data should be deposited in an appropriate database BEFORE acceptance.
2. The method to determine decay rates is certainly not state-of-the-art, and has been superseded by other approaches that do not require a transcriptional shut-off. However, it does have the great advantage of its simplicity, which is probably why it was chosen by the authors to allow the application of a complex method to the RNA samples. Therefore, I consider that the choice of methods is appropriate.
3. p1, Abstract: 'We systematically measure the molecular phenotypes of interleaved coding and noncoding transcriptional events'. I don't understand the sentence
4. p3, similar to previous sentence: 'we measured the turnover of interleaved transcriptional events along the yeast genome differentiating the contribution of coding and non-coding transcripts to the transcriptional outcome from a genic locus.' - Again, this is unclear until one has read the whole paper
5. p.4 - 'We obtained robust measurements of decay rates for approximately 21500 isoforms, mapping to 3600 annotated protein-coding and 210 annotated noncoding genes... We detected approximately 2900 protein-coding genes with at least two 3' isoforms mapping within the annotated translatable coding sequence (CDS) or the 3' UTR of the gene' I am confused by these numbers - is the term isoform always been used in the same way? Does it reflect to every single cleavage position, or to clusters of them? This should be clearly defined and stated.
6. p.4 - 'The measurements of decay rates for individual genes agreed reasonably well with previous per-gene measurements of decay rates obtained using microarrays in the same yeast strain and a similar experimental setup' - the correlation should be quantified
7. p.5-6 - how common are non-coding isoforms (with respect to the corresponding coding ones)? It would be good to have an indication of this figure, as this would show the relative contribution of these isoforms to overall turnover rates.
8. p.9 - The observation that isoform pairs separated by less than 5 nucleotides have different stabilities is very surprising (and interesting), as I would expect these isoforms to be formed from a single PAS. It would be good to have some more discussion on this phenomenon - do the authors have a model of how this could work from the mechanistic point of view?

1st Revision - authors' response

20 January 2014

Reviewer #1:

*The study by Gupta et al. investigates the influence of having transcript isoforms on transcript stability in the yeast *S. cerevisiae*. The main conclusion is that 3'-UTR heterogeneity can for some genes contribute to large differences in transcript stability and that in some cases this is mediated by differential interaction of 3'-UTRs with Puf3, an established regulator of mRNA stability. This is in part a confirmation of what is known/expected based on previous single gene and genome-wide transcript heterogeneity studies. The importance is the confirmation itself and the genome-wide scale of the study, that also allows an assessment of how widespread this type of regulation is. The data both serves as a demonstration that transcript isoforms can (in principle) contribute to phenotypes (indicating that this is not a particularly wide-spread phenomenon) and will also be useful for scientists wishing to investigate this for their particular favorite gene(s).*

The general strategy used to determine decay rates is probably the one that has been most widely used in the past (shifting an RNA polymerase II temperature sensitive strain to the non-permissive temperature). This approach is a lot more problematic than the authors present. In the first place the allele already causes slower growth at the permissive temperature. Second, the measured decay rates are a combination of two things: the decay itself plus the effects of shifting the cells from 24 to 37 degrees. Recent work using an almost completely non-invasive method based on in vivo labeling of RNA with 4sU/4tU has quite convincingly shown that the invasive methods (eg Cu-phen, rpb1-1) are accompanied by massive, general effects on RNA stability due to the stress itself. It is really surprising that the authors don't use this approach (applied in yeast by the Cramer lab in a series of three papers by first authors Sun, Schwalb and Miller, some of which has been cited here) which is much, much better than rpb1-1 and yields quite different half-lives! Although this has an enormous impact on the decay rates determined here (essentially this all needs

to be redone), this fortunately doesn't impact their general finding that different isoforms can influence stability.

The authors should however properly explain the difference between rpb1-1 determined decay rates and those most recently determined using in vivo labeling, pointing out the large difference in determined values and that the in vivo labeling methods are superior because they are not invasive. It is really important to place the current study in the context of what has been done before and not to give the impression that using rpb1-1 is a good idea - it is no longer.

We thank this reviewer for their thorough assessment of our study and the opportunity to address the metabolic labeling method (Dynamic Transcriptome Analysis method) implemented by the Cramer lab in several important studies of transcriptome dynamics. We agree that this method of metabolic labeling is less invasive than the *rpb1-1* approach; we thus would certainly have preferred to use it for this study and in fact began the project using metabolic labeling. Unfortunately, this application of the metabolic labeling technique presented significant analytical challenges that eventually made the *rpb1-1* method more appropriate for this study. In particular, the implementation of the statistical model used for the estimation of RNA turnover is dependent on estimating the labeling bias or the dependence of incorporation of 4sU/4tU with respect to the length of the transcript, as explained in the first manuscript detailing this method (Figure 2A from Miller et al., MSB 2011). Currently available methods for the quantification of polyadenylation isoforms by short-read sequencing do not allow the precise determination of full transcript length. While overlap with existing data (Pelechano et al., Nature 2013) allows a rough estimate of transcript length for some isoforms (in our case 65%), this is insufficient for reliable application of the statistical methods necessary for metabolic labelling protocols.

In addition, we have also observed that the current implementation of metabolic labeling in yeast has only been demonstrated for a low-resolution tiling array platform. As the structure of individual transcript isoforms, the focus of our study, cannot be resolved using tiling arrays, it was imperative for us to use sequencing-based methods. It would require considerable mathematical method development to adapt the metabolic labeling protocol to the sequencing platform (where issues like high dispersion might interfere with the labeling efficiency for lowly expressed shorter transcripts) to obtain proper estimates of turnover like those obtained by Miller et al. 2011 and Sun et al. 2012. Therefore, although we would have preferred to use metabolic labeling, the technical challenges described above led us to share Reviewer #3's conclusion that the simpler *rpb1-1* method was a more appropriate choice for our study. As requested by this reviewer, we have added a few lines in the discussion to raise these issues regarding metabolic labeling in the context of our study (Discussion: Paragraph 6: Line 11).

Figure 4 and the section describing these results. This figure should be accompanied by a statistical test (overall anova followed by pair-wise t test that is multiple testing corrected). The results of this test should also be clearly stated in the section describing this so that it is clear what the significance is of these different classes. If insignificant, this needs to be stated.

We thank the reviewer for the suggestion and have implemented it in the latest version of the manuscript. Changes have been made to Figure 4 and the text associated with it under in the Results section (Paragraph 5)

minor issues

page 4: "The decay rates for each independent replicate displayed good agreement (Spearman correlation 0.71, Figure 2C)". I would suggest removing the word "good" from this sentence. I guess it depends on what you are used to but a correlation of 0.71 between two replicates of the same measurement is actually quite bad.

We agree with the reviewer and remove the word "good" as suggested.

typos

"after 1h after" (pg 4 line9) "modulate" page 7 should be modulated

We have implemented these suggestions pertaining to minor issues and thank this reviewer for their detailed attention.

Reviewer #3:

Multiple studies have revealed high prevalence of alternative polyadenylation in every eukaryotic organism studied. Although numerous single genes studies have shown that transcripts with different cleavage and polyadenylation sites (PAS) behave differently, no genome-wide analysis has been performed to systematically analyse the behaviour of different isoforms. This manuscript addresses this issue by combining a high throughput method to identify PAS with a classic approach to measure RNA half-lives. This is a novel and clever approach to a very important question. Overall the paper is of good quality, well written and the data are carefully interpreted. I do not think that the choice of method for measuring RNA stability is the best (see below), but this is still very good piece of work. I only have a few minor comments and remarks:

1. I did not see any information on data deposition, all sequencing data should be deposited in an appropriate database BEFORE acceptance.

We agree and have corrected this error. We had in fact deposited the data but misplaced the ArrayExpress access code in the acknowledgements section. This has been now put in the data availability subsection in the materials and methods.

2. The method to determine decay rates is certainly not state-of-the-art, and has been superseded by other approaches that do not require a transcriptional shut-off. However, it does have the great advantage of its simplicity, which is probably why it was chosen by the authors to allow the application of a complex method to the RNA samples. Therefore, I consider that the choice of methods is appropriate.

We thank reviewer #3 for the comments and for understanding the use of rpb1-1 for estimating isoform decay rates. As explained in greater detail in the response to reviewer #1, we indeed chose this method because of its simplicity and the fact that no assumptions or measures about transcript length are required to calculate a decay rate.

3. p1, Abstract: 'We systematically measure the molecular phenotypes of interleaved coding and noncoding transcriptional events'. I don't understand the sentence

4. p3, similar to previous sentence: 'we measured the turnover of interleaved transcriptional events along the yeast genome differentiating the contribution of coding and non-coding transcripts to the transcriptional outcome from a genic locus.' - Again, this is unclear until one has read the whole paper

We used the word “interleaved” to highlight that we estimate the turnover of overlapping transcripts as shown in Figure 3. None of the current sequencing methods have calculated turnover of overlapping transcripts genome-wide. We have replaced word “interleaved” with the more appropriate word “overlapping” in the text and clarified that we are referring to multiple transcripts being produced from the same locus (Abstract: Line 4 and Introduction: Paragraph 4: Line 9).

5. p.4 - 'We obtained robust measurements of decay rates for approximately 21500 isoforms, mapping to 3600 annotated protein-coding and 210 annotated noncoding genes... We detected approximately 2900 protein-coding genes with at least two 3' isoforms mapping within the annotated translatable coding sequence (CDS) or the 3' UTR of the gene' I am confused by these numbers - is the term isoform always been used in the same way? Does it reflect to every single cleavage position, or to clusters of them? This should be clearly defined and stated.

We apologize for the confusion and have now clarified that we consistently use “isoforms” to describe single cleavage positions and not clusters (Results : Paragraph 3: Line1).

6. p.4 - 'The measurements of decay rates for individual genes agreed reasonably well with previous per-gene measurements of decay rates obtained using microarrays in the same yeast strain and a similar experimental setup' - the correlation should be quantified

The quantification of correlation between decay rates estimated per gene using both setups had been provided in supplementary figure S1C, and for clarity we now explicitly state these estimates in the main text (Results: Paragraph 3: Line12).

7. p.5-6 - how common are non-coding isoforms (with respect to the corresponding coding ones)? It would be good to have an indication of this figure, as this would show the relative contribution of these isoforms to overall turnover rates.

The median composition of coding isoforms per gene is approximately 86%; i.e. most genes have predominantly coding isoforms, making the contribution of non-coding isoforms to the overall turnover from a genic locus relatively small. We have added a Supplementary figure S3 and now cite this figure in the main text (Results: Paragraph 5: Line 17).

8. p.9 - The observation that isoform pairs separated by less than 5 nucleotides have different stabilities is very surprising (and interesting), as I would expect these isoforms to be formed from a single PAS. It would be good to have some more discussion on this phenomenon - do the authors have a model of how this could work from the mechanistic point of view?

We were also quite intrigued by this observation, and it will definitely be interesting to look into the mechanistic aspects of how changes of a few nucleotides in transcript length can impact its stability and how these changes are regulated. As requested by this reviewer, we have added a few lines in the discussion regarding this point (Discussion: Paragraph 1: Line 22).

2nd Editorial Decision

22 January 2014

Thank you again for submitting your revised work to Molecular Systems Biology. We are globally satisfied with the modifications made and we will be able to accept your paper for publication pending the following minor amendments:

1. We appreciate the explanations provided in the point-by-point letter with regard to the choice of methodology to investigate decay rates. Given that reviewer #1 made this comment: "The authors should however properly explain the difference between rpb1-1 determined decay rates and those most recently determined using in vivo labeling", we would kindly ask you to address this point in a more explicit manner in the Discussion section.

2. Sup Fig S3B, C: is it $\log_2(\text{fold change})$ that is shown? If yes, please correct the labeling of the X axis.

Please resubmit your revised manuscript online, with a covering letter listing amendments and responses to each point raised by the referees. Please resubmit the paper ****within one month**** and ideally as soon as possible. If we do not receive the revised manuscript within this time period, the file might be closed and any subsequent resubmission would be treated as a new manuscript. Please use the Manuscript Number (above) in all correspondence.

2nd Revision - authors' response

26 January 2014

Thank you once again for rapid decision and very efficient review.

As requested we have added a paragraph (in red) explaining the differences between rpb1-1 determined decay rates and those most recently determined using in vivo labeling. We have also corrected the labeling of Sup Fig S3B and attach here the extra text for the HTML version of the paper.

Please let us know if you require anything further.